# Shared decision-making and deprescribing to support anti-thrombotic therapy (dis)continuance for persons living with cancer in their last phase of life: A realist synthesis

Justin Jagosh[1,2], Mark Pearson[1*], Sarah Greenley[3], Anthony Maraveyas[1], Gamze Keser[1], Fliss E. M. Murtagh[1], Simon Noble[4], Michelle Edwards[4], Adrian Edwards[4], Anette Arbjerg Højen[5], Kathy Seddon[3], Frederikus A. Klok[6], Simon P. Mooijaart[7,8], Eric C. T. Geijteman[9], Miriam J. Johnson[1]

1 Wolfson Palliative Care Research Centre, Hull York Medical School, University of Hull, Hull, United Kingdom, 2 Centre for Advancement in Realist Evaluation and Synthesis (CARES), Vancouver, Canada, 3 Hull York Medical School, University of Hull, Hull, United Kingdom, 4 Division of Population Medicine, Cardiff University, Cardiff, United Kingdom, 5 Danish Center for Health Services Research, Aalborg University Hospital, Aalborg, Denmark, 6 Department of Medicine – Thrombosis and Hemostasis, Leiden University Medical Center, Leiden, the Netherlands, 7 Department of Internal Medicine, Section of Gerontology and Geriatrics, Leiden University Medical Center, Leiden, the Netherlands, 8 LUMC Center for Medicine for Older People, Leiden University Medical Center, Leiden, the Netherlands, 9 Department of Medical Oncology, Erasmus MC Cancer Institute, Rotterdam, the Netherlands

* mark.pearson@hyms.ac.uk

## Abstract

### Introduction

Patients with cancer are at increased risk of thrombotic complications from both the disease and its treatments, with antithrombotic therapy (ATT) usually continued in the last phase of life where the benefit is less clear and there is high risk of harms. Physiological changes toward the end of life increase the risk that ATT will cause serious bleeding events, but discussion between clinicians and patients of ATT risks and benefits is sub-optimal. This realist synthesis explores shared decision-making (SDM) to: (a) provide insights into why prescribing continues in end-of-life care; (b) build a conceptual platform for optimizing ATT prescribing for persons living with cancer towards end-of-life.

### Methods and findings

We conducted a realist synthesis using context-mechanism-outcome configurations and 'if…then' statements. A total of 17,036 citations identified across 10 databases (Medline, EMBASE, APA PsycInfo, CINAHL Complete, CDSR, CENTRAL, EPIS-TEMONIKOS, Web of Science Core Collection, Assia, Google Scholar). Ninety-one papers included following reverse chronology quota record screening (from: database searches (n = 56), consortium experts (n = 35)). Included papers: quantitative (n = 40),

**Data availability statement:** All relevant data are within the manuscript and its Supporting Information files.

**Funding:** The study is part of the research project SERENITY – "Towards Cancer Patient Empowerment for Optimal Use of Antithrombotic Therapy at the End of Life" (https://serenity-research.eu/). This project has received funding from the European Union's Horizon Europe research and innovation action under grant agreement No 101057292 (awardees: MP, MJJ, FEMM, FAK, SN). Additionally, United Kingdom Research and Innovation (UKRI) has provided funding under the United Kingdom government's Horizon Europe funding guarantee [grant agreement number 10039823 for Cardiff University and 10038000 for Hull York Medical School] (awardees: MP, MJJ, FEMM, SN). The funders had no role in study design, data collection and analysis, decision to publish, or preparation of the manuscript.

**Competing interests:** I have read the journal's policy and the authors of this manuscript have the following competing interests: FM is a UK National Institute for Health and Care Research (NIHR) Senior Investigator. The views expressed in this article are those of the author(s) and not necessarily those of the UK NIHR, or the UK Department of Health and Social Care. SN holds a Marie Curie Chair in Palliative Medicine.

**Abbreviations:** ATT, antithrombotic therapy; CMO, context-mechanism-outcome; PIMS, potentially inappropriate medications; PPI, patient and public involvement; RCQRS, reverse chronology quota record screening; SDM, shared decision-making.

qualitative ($n = 17$), mixed-methods ($n = 2$), evidence syntheses ($n = 16$), commentaries ($n = 9$), case reports ($n = 7$). Exclusion criteria: persons <18 years, non-English language, not focused on SDM or deprescribing in palliative care. An analytic appraisal journal was used with realist logic to synthesize insights from included papers (contents from 43/91 included in this paper). The concept of 'prescribing inertia' was used to formulate explanatory theories about clinician reluctance to deprescribe and the mechanisms underpinning SDM, including (a) the meaning medications have to end-of-life patients (e.g., 'life preserving' or 'symptom management') and public awareness of medications (e.g., high-profile chemotherapy versus low-profile ATT) are determinants of: (i) clinician motivation to engage patients around deprescribing, (ii) patient understanding, volition and participation in SDM; (b) SDM for ATT deprescribing requires sensitive engagement with patients and families without removing positivity around survival and continuing clinician interest in their welfare; (c) multi-disciplinary clinical decision-making about timing and suitability of deprescribing in end-of-life care requires specialized consensus-driven processes and evidence-based decision support tools; (d) if patients are healthy enough, empowerment interventions outside clinical encounters (e.g., health literacy apps) may increase patient and family readiness to engage in deprescribing conversations; (e) organizational investments can facilitate discussion of deprescribing (e.g., improved electronic medical record prompts, clinician communication skills training, data presentation to clinicians of actual ATT risks). Limitations: despite robust screening and selection, the sample of included papers does not reflect the entirety of eligible source material and did not include a systematic search for papers focusing on low- and middle-income country countries.

## Conclusion

Implementation of ATT deprescribing is enabled or constrained by (a) the meaning of medications to patients; (b) clinician engagement and understanding; (c) multi-disciplinary clinical decision-making processes (including support tools); (d) patient empowerment; (e) organizational investment. Addressing these multi-level factors, including the development of SDM tools, can address the prescribing inertia that may cause devastating impacts on patients and their families as well as moral distress amongst healthcare staff. This study was performed as part of the Horizon-Europe funded SERENITY project.

---

## Author summary

### Why was this study done?

- People with cancer are often prescribed blood-thinning medication because of the increased risk of blood clots from cancer treatments.

- However, blood-thinning medications have risks too, especially in late-stage cancers in which they may cause significant bleeding (rather than prevent blood clotting).

- Conversations about stopping blood-thinning medication can be very difficult, but may be helped by a better understanding of the importance of shared decision-making.

## What did the researchers do and find?

- We used an approach called 'realist synthesis' to better understand the underpinning mechanisms and contexts of shared decision-making relevant to conversations about stopping blood thinning. We used a wide range of sources, including empirical literature and input from healthcare staff and citizens.

- It is harder to stop than start a medication ('prescribing inertia') and conversations about stopping are affected by patients', families', and clinicians' knowledge and belief, as well as policies in the healthcare system.

- We organized the analysis of literature across three main areas: (1) identifying the "prescribing inertia" problem in end-of-life care; (2) Understanding the meaning of shared decision-making in end-of-life deprescribing; and (3) Examining the range of training approaches and decision support tools for patients, families and clinicians to foster shared decision-making in end-of-life care

- We found that there are many ways to foster conversations about deprescribing to promote goal concordance and quality of life for persons living with cancer in their last phase of life.

## What do these findings mean?

- Shared decision-making support tools can aid patients' and clinicians' understanding of risks and benefits of continuing or stopping blood-thinning medication.

- However, shared decision-making can be limited by the different approaches of healthcare staff, disempowerment of patients and families, and organizational factors.

- Our review has drawn on a wide range of research conducted primarily in high income countries and contributions from stakeholders in Europe (including people with lived experience). Findings suggest that deprescribing in palliative care is impacted by societal norms, organizational factors, and the quality of collaborative working amongst healthcare providers, patients and family members. Tailoring decision supports for anti-thrombotic medication can be supported by the framework and realist analysis presented in this paper.

## Introduction

Patients with cancer are at increased risk of venous and arterial thrombotic complications, both from the cancer as well as its treatments [1]. Antithrombotic therapy (ATT) is usually continued for patients with cancer in the last phase of life where the benefit is less clear, and there is high risk of harms. [2,3]. Although ATT is effective in reducing the risks associated with cancer-related thrombosis, physiological changes toward the end of life increase the risk that ATT will cause serious bleeding events [1]. ATT drug interactions and pill burden provide further disadvantage. These changes and considerations are not always factored in considering the risks and benefits of ATT prescribing over the course of treatment [4].

Engaging patients and families in conversations around end-of-life deprescribing, including ATT, is warranted [5]. Upheaval caused by a terminal diagnosis, as well as patient beliefs and values around death and their assumptions about clinical care interact in complex ways with clinician opinion of the risk of ATT continuation or cessation [6,7]. Clinicians

may feel reluctant to converse about deprescribing and worry about distressing patients and their families by triggering feelings of hopelessness around survival. As a result, ATT is often continued until death or until a serious bleeding event, the latter creating traumatic burden for patients and their families already experiencing an emotionally challenging period. Tools to help clinicians with persistent uncertainty about initiating deprescribing conversations and shared decision-making (SDM) are sparse [8]. One tool to help antithrombotic decision-making in advanced disease used in a retrospective study of 111 people with advanced cancer and indicated the tool was safe, [9–11] but there are no data from larger cohorts.

To address this complex clinical problem, a five-year Horizon-EU study called: "Towards Cancer Patient Empowerment for Optimal Use of Antithrombotic Therapy at the End of Life" (SERENITY) [12–14] was funded to develop and test a SDM support tool for clinicians and patients to improve the optimization of communication on the benefits and harms of ATT and possible deprescribing with patients living with cancer in the last phase of life (Horizon Europe 101057292). This paper reflects the work of the first phase of the research project, in which a realist synthesis [15,16] was undertaken to extrapolate insights from a broad set of literature in SDM, deprescribing and palliative care. The scope of the review was intentionally kept broad (a) because there was a lack of studies specific to using SDM for ATT deprescribing and (b) to extrapolate creative insights from diverse areas to inform our specific understanding related to the SERENITY research agenda. Table 1 presents the review questions:

## Methodology and methods

The review used realist methodology, which is a theory-driven approach that seeks to uncover explanatory mechanisms underpinning interventions and the key elements of the context that support or impede the mechanisms [15,16]. Table 2 provides definitions of key terms used in realist methodology as well as in the study area:

To accomplish the review, Pawson's five iterative stages for realist synthesis were undertaken [26] including: (1) Developing initial programme theories; (2) Searching for evidence to refine theories; (3) Selecting sources to include; (4) Extracting and organizing evidence; and (5) Synthesizing the evidence.

### (1) Developing initial programme theories

An initial sample of papers ($n = 20$) was retrieved from the larger database search to explore content and draft initial programme theories. These theories took the form of 15 "if…then" statements capturing causal insights regarding deprescribing in the last phase of life and SDM in that context. In understanding how context impacts the mechanisms associated with optimal deprescribing, this early work provided programme theories as groundwork for subsequent analysis. The list of initial programme theories is found in the S1 File.

### (2) Searching for evidence to refine the theories

Initial scoping searches were conducted by an information specialist (SG) to identify key terms and quality of available evidence. Search terms to address the review questions were identified from the indexing of known relevant studies and using the PubMed PubReminer tool to examine focused search results. Existing validated search filters for palliative care [27,28], and related systematic reviews on deprescribing in palliative patients living with cancer [29–31] were also

**Table 1. SERENITY realist synthesis questions.**

| |
|---|
| 1. What are the processes by which deprescribing decisions are taken when caring for adults with long-term conditions and at the end of life? |
| 2. In relation to these deprescribing decisions: a. What mechanisms are believed to operate at different levels (individual, family, team, professional, organizational) that may explain why intended and unintended outcomes occur? b. How do different contexts impact on the operation of these mechanisms? |
| 3. How can knowledge of context-mechanism-outcome configurations inform the design of a shared decision-making tool for optimal prescribing of antithrombotic therapy for patients with cancer in their last phase of life? |

**Table 2. SERENITY study definition of terms for realist methodology and the clinical area.**

**Realist synthesis methodology**: A methodology for literature review that seeks to understand how, for whom, and in which circumstances programmes and services work to achieve outcomes [15]. Realist synthesis uses the 'context-mechanism-outcome' (CMO) configuration as a heuristic for theory development and data analysis [16].

**Realist programme theory (e.g., 'if…then')**: An explanatory statement detailing how a programme or programme component works. Initial programme theories are often articulated as 'if…then' statements reflecting partial, rough CMO thinking which are then tested against evidence.

**Programme mechanism**: The underpinning generative force that explains how outcomes manifest [17,18] from programmatic efforts. Specifically, it is how stakeholders involved in programmes (e.g., patients, clinicians) respond/react to resources [15]. Mechanisms for ATT deprescribing in end-of-life care can involve clinician, patient and family realizations for the importance of clinical risk-benefit considerations in relation to patient preferences. Confidence and motivation to deprescribe are other related mechanisms. Outcomes include actual medication optimization and resultant changes in quality of life for patients.

**Context**: Aspects of the environment of a programme or service that impact or trigger the mechanisms of that service [19,20]. For ATT deprescribing, this includes the society-wide awareness profile of ATT, cultural attitudes about the role of clinicians, religious beliefs on medical treatments at the end of life, as well as clinical and organizational factors such as patient case loads for clinicians, payment schemes and clinical leadership attitudes and priorities in relation to supporting deprescribing efforts.

**Deprescribing**: The process of tapering, stopping, discontinuing, or withdrawing drugs, with the goal of managing polypharmacy and improving outcomes [21].

**Shared decision-making (SDM) (in clinical contexts)**: Communication by which clinicians introduce choice regarding medical treatments, describe the benefits and drawbacks, and help patients and families explore these options in relation to their values and preferences to arrive at informed decision-making [22]. SDM may help patients develop informed preferences [23].

**Professional equipoise (in clinical contexts)**: An attitude and comportment held by clinicians that presents a neutrality of opinion when multiple valid treatment options exist which benefit from patient and family input. Clinicians may list options that are reasonably available, including the option of not taking action, without displaying a preference for one treatment over another [24,25].

**PIMs list**: A decision support tool for clinicians to assess medication appropriateness in the care of older people and typically with multimorbidity. PIMs stands for 'potentially inappropriate medications'.

**EMR**: Electronic medical records are electronic files that clinicians use to store and share patient medical information.

**"Last Phase of Life" and "End of Life"**: There is no consensus in the literature on the use of these terms and source papers included in this review vary in their definitions. For this review, the terms are defined as follows: The last phase of life is estimated to be within 12 months before dying, in which a person is living with an eventually fatal condition, even if the prognosis is ambiguous or unknown. End of life is the phase of life roughly three months before dying in which a person is living with a fatal condition in which the death prognosis is fairly certain. End of life is different from active dying, which is usually given a timescale of a few days before death.

examined to develop an initial search string. This was tested and revised against a set of 47 known relevant results in the Medline database via OVID following feedback from the wider review team. After an initial strategy was developed focusing on deprescribing AND palliative care AND (shared decision-making or goal concordant care), a team decision was made to widen the main search to avoid missing studies on SDM/goal concordant care outside of deprescribing. The final main search structure was: (shared decision-making OR goal concordant care OR deprescribing) AND palliative care. Full search strategies, a search narrative and PRISMA-S checklist are shown in S2 File.

The formal search, screening and analysis of papers was conducted in two stages with the first stage having two parts (Step A and Step B) as depicted in Fig 1. Step A of stage one prioritized papers broadly focused on SDM and deprescribing in end-of-life care. Preliminary findings from this step were presented to stakeholders including a clinician consortium as well as a patient and public involvement (PPI) group. Step B of stage one followed from this consultative work and prioritized specific questions as identified by stakeholders about the nature of thrombosis and ATT deprescribing, organizational factors affecting deprescribing, and moral distress of healthcare staff in the palliative context.

Both steps A and B of stage one involved a broad, world-wide search of multiple databases: Medline All, Embase, APA PsycINFO (all via OVID), CINAHL via Ebsco, The Cochrane Library of Systematic Reviews and Cochrane Central Register of Controlled Trials via The Cochrane Library, ASSIA via Proquest, SCI-EXPANDED, SSCI, AHCI, CPCI-S, CPCI-SSH, ESCI via Web of Science, Epistemonikos and Google Scholar. Searches were completed between 17 and 18 October 2022 and in order to ensure manageability and relevancy to contemporaneous contexts, the date range was limited

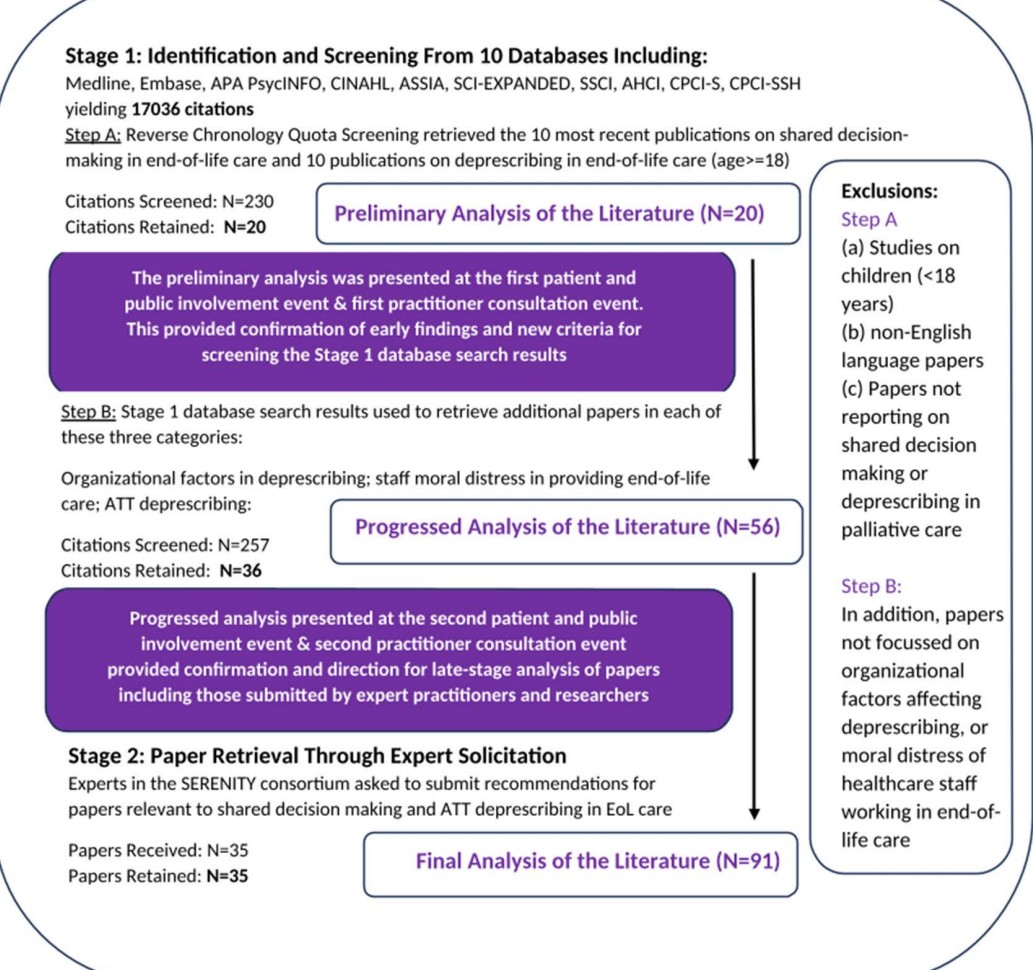

**Fig 1. Identification and screening flow chart.**

to results post 2010. Results were imported into an Endnote library for duplicate removal before being uploaded into RAYYAN citation management software [32], which was used to house the retrieved citations. Stage 2 involved soliciting relevant papers from SERENITY consortium stakeholders.

## (3) Selecting sources to include

Stage 1 (Step A) Inclusion/exclusion criteria

Literature was searched for evidence that describes and assesses interventions to (a) improve SDM and deprescription efforts with adults (age ≥ 18 years) towards the end of life; and (b) describing barriers to goal concordance and deprescription.

Exclusions included (a) studies involving children; (b) non-English language papers; (c) papers not reporting on SDM or deprescribing in end-of-life care

Stage 1 (Step B) Inclusion/exclusion criteria

The step B search was conducted to identify bodies of literature to build upon or refine the theories identified and developed in Step A, focusing on the use of anti-thrombotic medication in patients living with cancer at the end of life. Papers on how organizational factors impact deprescribing efforts and moral distress of healthcare staff working in end-of-life care were selected through stakeholder consultation Targeted searching was conducted within remaining database search results for these specific areas. Exclusions included (a) studies involving persons under 18 years of age; non-English language papers; (b) papers not focused on ATT in patients with cancer at the end of life, not focussed on organizational factors affecting deprescribing, or moral distress of healthcare staff working in end-of-life care.

Reverse Chronology Quota Record Screening (RCQRS) [33] was used for both Step A and B in Stage 1. RCQRS is a screening strategy in which established *a priori* quotas are used to screen database records starting from the most recent date and working in reverse chronology until a quota is filled. Five quota categories with a rough quota limit of 10 papers for each, comprised the screening stage. For Step A, this included deprescribing in palliative care and shared decision-making in palliative care. For Step B this included healthcare staff moral distress in palliative care; organizational factors affecting deprescribing; and papers on ATT. Types of data sources for Stage 1 and 2 screening included: 1. Empirical papers, commentaries, editorials on SDM in end-of-life care (age ≥ 18 years); 2. Empirical papers, literature reviews, commentaries, editorials on deprescribing processes with adults (age ≥ 18 years) towards the end of life. Two members of the team (JJ) and (GK) independently screened papers using RAYYAN, a software screening tool [32]. A third member (MP) was involved in resolving disagreements. Stage 2 solicited papers by experts in the SERENITY stakeholder advisory groups to achieve the final set of included studies. See Fig 1 for a visual depiction of the identification and screening process.

### (4) Extracting and organizing the evidence

The analysis involved the use of an analytic appraisal journal to support team-based extractions of critical insight from papers. These insights were then organized across key theory areas mapped to the service architecture in end-of-life care. Papers were read with an eye to understanding causal mechanisms and contexts. These were then explored through explanatory writing in the appraisal journal using selected quotes from papers and building CMO configurations.

### (5) Synthesizing the evidence

The journal's content was reviewed by the core team and commented on for relevance and sense checking. The realist logic of analysis and appraisal journaling, in tandem with the initial programme theories developed in phase 1 were used to consolidate top-level 'if…then' statements (see Table 3) and context-mechanism-outcome (CMO) configurations (see findings section below) The analytic process utilized the concepts of evidence juxtaposition, reconciliation, adjudication, and consolidation and followed the RAMESES reporting standards for realist synthesis [34] (see S5 File).

### Stakeholders consultation

Four stakeholder meetings were held across the 12 months of the review to ensure relevance of the review questions and to identify any gaps as the data analysis progressed. Two of these were with the SERENITY Consortium, comprising researchers and clinicians from 14 hospitals located in eight European countries including Italy, Spain, Denmark, the Netherlands, France, United Kingdom, Germany and Poland, with specializations in oncology, cardiology, geriatrics, hemostasis, hematology, family medicine, palliative care, health economy, epidemiology, health communication and psychology.

The other two meetings were held through a patient and public involvement group, comprising people living with, and supporting people with cancer, as well as members of the public at large. The stakeholders at these meetings were presented with the research questions and the on-going analysis as it progressed over time. They were asked to respond to the analysis with confirmation, disconfirmation and adding new theories and areas of importance. This process provided

**Table 3. Top-level "If…Then" statements conveying key insights from the literature.**

| |
|---|
| **1. The "prescribing inertia" problem in end-of-life care:** |
| **1a. Socio-cultural factors causing "prescribing inertia" for patients in the last phase of life:** |
| *If societal or cultural norms assume that the clinician should exclusively guide pharmaco-therapeutic decisions, and if ATT has a low profile of awareness in the public domain, then clinicians may feel reluctant to engage in shared decision-making with patients and families. Clinicians may feel concerned that patients and families will perceive their gestures toward SDM for ATT deprescribing as lacking needed leadership and clinical expertise for complex medical decision-making.* |
| **1b. Organizational barriers causing ATT "prescribing inertia" in the last phase of life:** |
| *If hospital leaders do not endorse and communicate endorsement of the need for ATT deprescribing activity, clinical staff will feel reluctant to engage in SDM about ATT deprescribing with patients. Clinicians may be confused about the appropriateness of communicating ATT deprescribing to patients.* |
| **1c. Clinical barriers causing ATT "prescribing inertia" in the last phase of life:** |
| *If clinicians feel uncertain about the benefits vs. drawbacks of anti-thrombotic therapies for individual patient needs, worry about causing harms by removing ATT, and worry about upsetting patients with communication around ATT deprescribing, then prescribing inertia will persist despite the potential benefits of deprescribing.* |
| **2. Shared decision-making in end-of-life deprescribing** |
| **2a. Clinician-patient-family shared decision-making for end-of-life deprescribing** |
| *If clinicians maintain professional equipoise in the context of profound existential uncertainty, then they will offer guidance on benefits and risks of ATT for individual patient circumstances to simulate and give prominence to the patient voice rather than obfuscate of patient preferences. Genuine shared decision-making between clinicians, patients and families will leave open the possibility for clinical expertise to influence, and in some cases, override patient preferences in the event patients and/or families indicate their desire for preventive but futile treatments including the risky continuation of ATT treatment.* |
| **2b. Interprofessional shared decision-making for end-of-life deprescribing** |
| *If a collaborative model for ATT deprescribing decision-making fosters trust across clinician team members, despite differing opinions, then clinicians will feel supported in their efforts to explore deprescribing and will not fear offending colleagues or original prescribers who may not agree with arguments for ATT removal. Such a collaborative model will increase consensus building and reflection on the deprescribing process.* |
| **3. Education, training and decision support for end-of-life shared decision making** |
| **3a. Clinician education, training and decision support to prepare for end-of-life ATT deprescribing** |
| *If clinicians are provided with appropriate education and training to understand the meaning of professional equipoise in the context of clinical uncertainty, they will feel confident to initiate conversations with patients on ATT deprescribing, at the right time and with an understanding of balancing patient preferences and values with clinical expertise.* |
| **3b. Patient and family health literacy and decision support to prepare for end-of-life deprescribing** |
| *If patients and families are provided with appropriate information, tools and resources on ATT risks and benefits for cancer care toward the end of life, then they will understand the opportunity to become active co-agents in decisions about their care and will be ready to engage in a shared approach to deciding on medication continuation, reduction or discontinuation.* |

direction and validation to the developing analysis, theoretical framing of the review, and developing search terms for the second phase of searching the literature. See S3 File for further detail regarding these meetings.

A protocol for the review is registered in the international prospective register of systematic reviews (PROSPERO: CRD42022375000).

## Results

### Findings

The main search yielded 17,036 citations after duplicates were removed. The selection of studies in Stage 1, S involved screening of 230 articles yielding 10 papers on deprescribing and 10 papers on shared decision-making in palliative care. The selections of studies in Step B included an additional 257 papers screened after consultation with stakeholders. This yielded 10 papers on moral distress in end-of-life care, 7 papers on organizational factors influencing deprescribing efforts and 19 general papers deprescribing ATT. In Stage 2, 35 papers were retained through recommendations by consortium of experts linked to the wider SERENITY project. Of the 91 papers retained, contents from 43 papers were used and cited in the findings section below. See S4 File for a description of studies including areas of investigation, country and year published.

The findings of the literature review are organized across three areas and seven sub-areas (Fig 2). For each sub-area, a top-level programme theory is provided in Table 3 and numerous CMO configurations embedded in the analysis. The main mechanisms of SDM include: (a) clinician awareness for the need to engage in deprescribing conversations with patients; (b) clinician motivation to engage in deprescribing conversations with patients; (c) patient and family understanding of clinicians' intent in initiating deprescribing conversations; (d) patient and family motivation to engage in a co-decision-making process for deprescribing. The contexts include the enabling and disabling factors at societal, organizational and clinical levels that trigger these mechanisms whereas outcomes include optimal deprescribing, improved quality of life for people living with cancer in their last phase of life, and improved job satisfaction and reduced moral injury of healthcare staff who may otherwise witness severely distressing and disruptive bleeding events caused by a lack of ATT deprescribing.

**The "prescribing inertia" problem in end-of-life care**

1a. Socio-cultural factors causing "prescribing inertia" for patients in the last year of life

1b. Clinical barriers causing "prescribing inertia" in the last year of life

1c. Organizational barriers causing "prescribing inertia" in the last year of life.

**Shared decision-making in end-of-life deprescribing**

2a. Clinician-patient-family shared decision making for deprescribing in the last year of life

2b. Interprofessional shared decision making for deprescribing in the last year of life

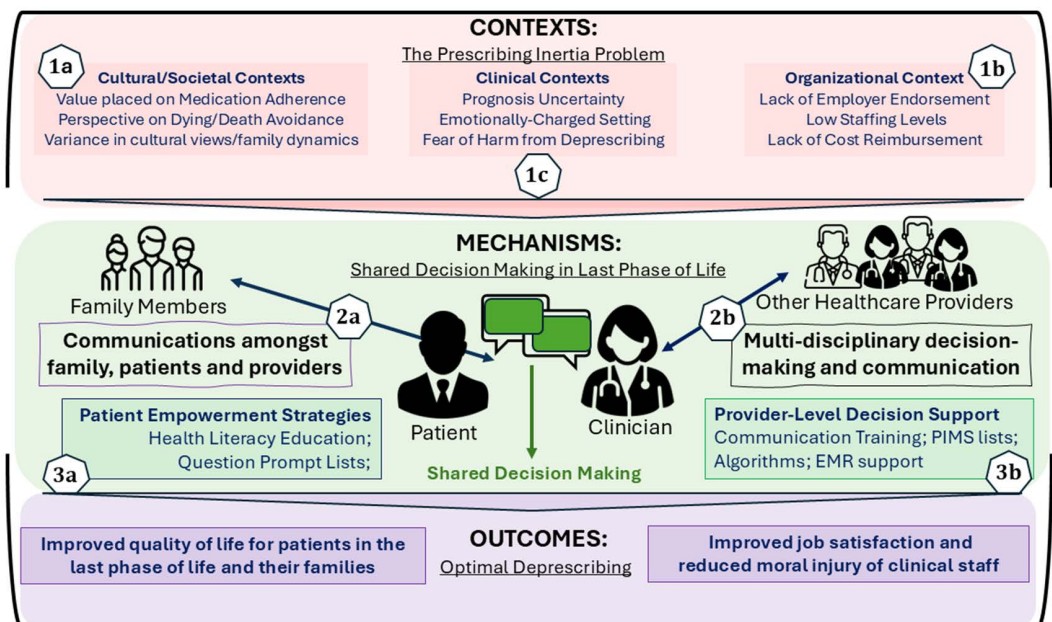

**Fig 2. Visual representation of theory areas for shared decision making and ATT deprescribing in end-of-life care.** EMR, electronic medical records; PIMS, potentially inappropriate medicines. Images designed by HANIS (Female Doctor #1, from www.flaticon.com), Freepik (Male Doctor #2, from www.flaticon.com), Gregor Cresnar (Patient, from www.flaticon.com), and Freepik (Family Members, from www.flaticon.com).

### Education, training and decision support for end-of-life shared decision making

3a. Clinician education, training and decision support to for end-of-life deprescribing

3b. Patient and family health literacy and decision support for end-of-life deprescribing.

### The "prescribing inertia" problem in end-of-life care

An important concept in the literature, namely prescribing inertia, has been captured in this review to identify numerous barriers in the context preventing the mechanisms of deprescribing from activating. Prescribing inertia means clinical inaction regarding medication review and SDM with patients about the use of potentially inappropriate medications (PIMS) [35]. Numerous factors are involved including lack of time [36], "profound existential uncertainty" in which clinicians are uncertain about the survivability of patients [8], not in the least due to the introduction of new systemic anticancer treatments [37], fears of adverse events arising from the removal of medications [38], preventing moral distress in staff [39] and pressures from a healthcare culture of medication adherence, organizational policies and societal norms. For example:

> "First, having spent years encouraging people to take preventive medicines, some prescribers find it challenging to suggest these are no longer necessary. Second, it is sometimes challenging to prognosticate that a person is in the last year or a few months of life. Third, discussing end-of-life care is inherently challenging due to the sensitive and emotional nature of such conversations" [40] (p. 4).

Despite the importance of deprescribing for reducing serious side-effects and pill burden [38] patients and clinicians can have unfounded optimism about prognosis which further contributes to prescribing inertia. In this social context, clinicians may feel uneasy about raising questions of medication withdrawal out of concern that patients and families would perceive such efforts as giving up, communicating imminent demise, or even bringing about death [40]. In addition, the ability of clinicians to recognize the signs of a reduction in expected survival and timely communication has been identified as critical to countering prescribing inertia and engaging in proactive medication review [41].

**CMO #1:** The culture of medication adherence, uncertainty about patient survivability, and the sensitive emotional state of a person with terminal cancer are significant barriers to deprescribing medications (context). Specialized efforts, supports and communication approaches can foster HCP confidence in initiating deprescribing conversations with patients (mechanism), thereby minimizing the chances of severely distressing and disruptive bleeding events for patients at the end of life (outcome).

Prognostication involving algorithm-based decision support tools can provide data to counter prescribing inertia to an extent. However, these are considered controversial by some, due to the fact they are always only partially accurate against unique patient circumstances which may supersede algorithm-based recommendations. Discrepancies between deprescribing suggestions provided through prognostic modelling in the absence of factoring for unique patient characteristics may lead to clinical confusion and result in prescribing inertia:

> "recognizing when an older person is approaching end of life is a key challenge for physicians. Prognostic models, which are generally derived from large population-based databases, synthesise patient- and disease-related information to produce prognostic estimates. These estimates indicate the mortality risk for an average patient with a given set of risk factors under average circumstances. Relevant, specific information related to the individual patient, may not be included in the prognostic model, and, therefore, it is questionable whether prognostic models should be used to influence important decisions at an individual patient level." [42] (p. 467)

**CMO #2:** Given the complexity of individual patient circumstances (context), prognostic modelling based on averages may create confusion for healthcare providers (mechanism). This in turn may lead healthcare providers to abandon prognostic modelling and associated deprescribing efforts (outcome)

Positive attitudes and perceptions toward advance care planning by clinical staff do not guarantee more end-of-life conversations with patients, due in part to barriers that exist at organizational levels [43]. In addition, heavy workloads and administrative duties along with unsupportive reimbursement schemes have been identified as organizational factors impeding deprescribing [36]. In addition, healthcare policy may restrict who across healthcare teams has the authority to formalize goals of care documentation [44], which may impact SDM opportunities. Similarly the extent to which SDM processes are standardized [45] may positively or negatively impact the way clinicians engage with patients to overcome prescribing inertia.

**CMO #3:** Organizational factors such as time and administrative burden, along with policies on scope of practice for HCPs may exacerbate prescribing inertia (context). Supportive organizational policies which expand roles and offer staff the recognition for taking time to engage in SDM for deprescribing can improve motivation of staff to engage in medication review (mechanism), to safeguard against adverse events triggered by futile treatments (outcome)

Cultural norms may also impact deprescribing efforts [46] with differing views on the value of active treatment for life-limiting conditions. For example, differences in the concept of patient autonomy in western and non-western cultures [47] may impact SDM with the fact that "many non-Western cultures value the inclusion of family in end-of-life decision-making over patient autonomy" (p. 9). Some cultural and religious norms also dictate the continuation of therapies until death based on the belief that suffering is an important part of preparing for the afterlife. Some patients may also expect to take a passive role and defer decisions to the clinician, especially at the initial stage of a terminal prognosis:

> *"research has reported that newly diagnosed patients chose to play a passive role in the decision-making and preferred their physician to play a paternalistic role"* [46]

In some cases, patients prefer not to know their prognosis, making SDM a limited activity [48] and in other cases, families relay clinical advice to their religious or spiritual guides who appraise the information according to religious beliefs and provide direction accordingly [49]. Given such diversity of opinions and values in multi-cultural societies, and legal frameworks that dictate the scope of practice for withdrawing and withholding life extending treatments, organizational and clinical leaders may convey messages to staff to do 'what the family wants' to avoid conflict and litigation [50]. Such actions may increase moral distress in staff if they witness patients suffering unduly in their opinion, due to family and religious beliefs dictating that treatments should be maintained to the end of life. The legal framework for life-sustaining treatment in a country will also determine the dynamics of end-of-life conversations with patients and families. For example, moral distress may also be created from legal frameworks in some countries that may not permit physicians to remove curative treatments against their better clinical judgement [51].

Although patients and families should be given autonomy in making informed decisions around the risks of aggressive treatments, clinicians may experience moral distress when patients insist on treatment that the clinicians deem futile [52,53]. Poor outcomes from failed attempts at SDM may lead to maintaining paternalistic approaches in the future and a re-enforcement of prescribing inertia [54]. Countering moral distress with ethical perspectives can reduce prescribing inertia, supporting clinicians in their realization that there is not necessarily one "right" way to do things, and that considering the mix of factors can lead to a conclusion that best preserves the practitioner's integrity and also serves the patient in the best way possible given difficult realities associated with end-of-life care [55].

**CMO #4:** Patients have different preferences regarding active treatment in their last phase of life (context), possibly informed by religious or cultural norms. Ethical perspectives which prioritize SDM and which acknowledge that some patients would prefer the risks of active treatment even if those risks lead to adverse events, will increase confidence of HCPs to overcome prescribing inertia motivation to engage in deprescribing conversations with patients (mechanisms) while minimizing moral distress that arises from goal discordance between providers and patients (outcome)

### Shared decision making in end-of-life deprescribing

The literature indicates that SDM between clinicians, patients and families entails an approach beyond the simple asking of what patients and families prefer or value regarding decisions for end-of-life care [8]. Rather, SDM in the end-of-life context requires that patients and families are invited to participate in conversation about medical options by being provided adequate, digestible information, and that clinicians have 'professional equipoise' [24] (p. 2) to not unduly influence decisions at the expense of patient preferences. Success in SDM has been defined as when all parties (clinician, patient, and family) depart from SDM conversations having been influenced by the other [23].

Clinicians may need to dispel their preconception that patients are incapable of engaging in SDM which is precipitated by patients not having enough information to voice preferences. Some authors have described the need for broadening the scope of communication with patients to allow patients to feel comfortable expressing personal values and beliefs around care needs:

*GOC [Goals of Care] discussions are increasingly recognized as mattering less about patients' specific preferences for care at the very end of life and more broadly about their personal values, perspectives, and belief systems"* [56] *(p. 3400).*

The literature indicates that patients typically prefer clinicians to provide an honest expert perspective on their prognosis and advice about how to mentally prepare for the eventuality of death, even when they are still relatively healthy and stable:

*[the research demonstrated] that very few patients (<5%) would lose all hope for survival if they were told there was a high chance they would die soon. Most respondents reported that if their health was deteriorating but they were still receiving care of curative or life-prolonging intent, they would be comfortable discussing the potential for death so that their wishes could be known.* [57] *(p. 1395)*

Clinicians may fear upsetting patients or exposing genuine uncertainty about medical decisions [46]. Shock, denial and rumination may also distract or deter patients from conversations about treatment plans with clinicians. Upon initial diagnosis, patients may be inclined to give decision-making power to clinicians in the hope that expert medical intervention will save or significantly prolong their life. However, over time as an illness progresses and patients accept that they are indeed moving toward the end-stage, they may become more focussed on comfort, with time increase their health literacy, and become more willing to engage in decisions where the primary focus is on optimizing quality of life.

However, clinicians may perceive patients' initial reluctance to engage in SDM as an indication that they lack interest or the ability. Clinicians may maintain this assumption throughout the illness journey, described by some as 'belief inertia.' In these instances, clinician beliefs about patients' preferences are set in initial consultations and not re-assessed over time. This complexity leaves clinicians uncertain about the appropriate timing of SDM, compounded by variation in patient preferences about their level of involvement in decision-making. Some patients do not want to be involved in treatment decisions at any stage in the disease progression and clinicians may avoid conversations from such unknowns:

*"while some people with hematological malignancies wanted open, honest prognostic information, others did not want specific details of their prognosis. This discrepancy of opinion affects clinicians' confidence in the timing of end-of life discussions and patient preparation"* [57] *p. 1387.*

**CMO #5:** Differences in patient preference regarding information on their prognosis and their early deferral to clinician expertise (context) affects clinicians' assumptions on patient capacity for SDM and sense of confidence in initiating end-of-life discussions (-mechanism), resulting in a lack of SDM around treatment in the last phase of life (outcome)

Evidence also suggests that clinician communication skills around deprescribing do not simply improve with practice [41] and that in the effort to engage patients in SDM, the delivery of complex medical information can be overwhelming. The emotionally charged context surrounding such discussions can reduce cognitive processing of information by patients and families. As a result, clinicians may communicate with an inflated tone of optimism to not upset or overwhelm patients:

> *many clinicians make strenuous efforts to provide information about these complex agents that is barely digestible. At the same time, the inherent uncertainty clouds the prognosis that death could be close, although impossible to predict. Euphemistic language often amplifies optimism. Rational comprehension is undermined by fear and emotion and so decisional capacity is severely compromised.* [57] *(p. 3)*

**CMO #6:** The emotionally charged environment of end-of-life care makes it difficult for patients and families to absorb and digest important medical information including medication review (context). As a result, clinicians may feel a responsibility to communicate optimistically (-mechanism) resulting in reduced SDM for deprescribing (outcome)

Use of specific language by clinicians, adopted through communication skills training, can reduce non-collaborative patient 'nudging', either toward sustaining the prescribing *status quo* or for deprescribing. This problem was referred to as 'persuasion over deliberation' [58] in which either active treatment is communicated as the default course of action, or else negative language is used to convey prognosis:

> *As a central component of shared decision-making, evidence suggests clinicians need to use specific language to avoid the risk that patients will feel abandoned or that clinicians are "giving up." Relating deprescribing of certain medications to their treatment goals and using positive or motivational language such as "optimize," "individualize," "maximize benefit and minimize harm," and "reduce pill burden" rather than negative language such as "quitting" and "stopping" can promote a process of shared decision-making with patients and their families.* [59] *(p. 694).*

One paper reflected on the debate about the suitability of nudging patients, and the role of nudging in SDM communication. Clinicians in this sense may communicate what other patients do in similar situations: *most patients would not choose to be resuscitated in your situation*. This may help patients gain perspective on typical courses of action but may also suppress an exploration of personal values and desires for care:

> *"there is a long-standing divide between those who stand against the use of nudges and those in (relative) favor of them. Those opposed argue that nudges, including those which encourage social comparison may alter patients' reasoning, reduce their voluntariness and threaten full patient autonomy. In contrast, there are others that consider nudges to be potentially ethically acceptable as they aim for decisions that have the patient's well being in mind, as long as they are not misrepresentations (especially important in the case of reference to norms and social comparison), and that the power differentials between the messenger and the person nudged are not too great'* [24] (p. 10).

The literature also suggests that clinician-patient communication requires that clinicians identify important opportunities in the dialogue to inform patients on the benefits and risks of maintaining active treatment at the end of life. For example, when patients ask their clinicians to 'do everything they can', clinicians have an important opportunity to clarify how active

treatment based on an aggressive approach to saving the life may come at increased health risks and possible reduced comfort in the last year of life:

> *All physicians reported interpreting requests to "do everything" as a "red flag", a sign to more thoroughly explore what "everything" meant to the patient or family, a time-consuming but necessary strategy to prevent future conflict … physicians stated that they often started discussions with the family that "doing everything" might inflict additional pain and suffering on a patient who would die regardless of the requested "heroic" treatment.* [60] *(p. 6)*

**CMO #7:** Discussions about treatment in end-of-life care can be fraught with uncertainty and fear (context). Clinicians can use specific language to foster collaboration, including careful, compassionate nudges to help patients realize the importance of comfort at a time when they may still be coming to terms with their prognosis (mechanism) leading to improved SDM for deprescribing and improved quality of life in the last phase of life (outcome).

SDM across interprofessional staff teams has been shown to counter prescribing inertia. For example, improved communication channels between hospital and other health services such as primary care physicians and pharmacists can lead to reduction of confusion about why medications are being deprescribed [36] and conflicting opinions that default to not taking action for deprescribing. Without a teams-based approach, clinicians can be reluctant to stop medications that fall outside their scope of specialty and would prefer to defer decisions to the original prescriber, unknown to them, who may be more familiar with the medication [61]. Lack of interdisciplinary working can lead to deprescribing hesitancy for fear of offending the original prescriber or feeling the need to defer to their expert judgement on the decision [36]. Alternatively original prescribers whose medications were prescribed for long-term chronic health conditions, and who are not sensitive to end-of-life care needs may default to advising against deprescribing unless informed to consider a palliative care perspective on need. Shared clinical record systems have been suggested to mitigate these barriers in the event there is not a structure in place for genuine multi-disciplinary team working [62].

The question has been raised regarding who best to initiate and engage in sensitive conversations with patients about end-of-life deprescribing, if not physicians. Nurses and 'health navigators' have been identified as beneficial in this regard [63] due to a lesser power differential with patients, more time and increased opportunity to engage in holistic conversation with patients and families. Additionally, given how staff may be assessing illness progression differently for a given patient based on their expertise and relationship to the patient, strategies have been developed within multi-professional collaborations that allow for uncertainty and difference of opinion to be heard to support consensus building. This is particularly important when multiple healthcare staff are caring for one patient but disagree on course of medical treatment for end-of-life care. One consultant specialist remarked:

> *"I would not withdraw [life sustaining intervention] without consensus of opinion and of at least two colleagues. I always seek the other opinions. Where another consultant thinks we are doing the wrong thing by initiating end-of-life care too soon, that person would have a veto for a while. The rest of us would say, "right, let's give it another twenty-four hours." So if you felt strongly, then your voice is heard over others until you change your mind, or they change their mind* [64] *(p.187).*

Alternatively, adopting a default-to-deprescribing approach whereby medications are deprescribed as a standard procedure may be inappropriate and increase conflict across multi-specialist teams working together to serve patients. For example, some specialists may feel that that medications important for both survival and quality of life have been unduly stopped when other staff express concern over potential side effects of continuation [65]. Managing conflicting opinions across the healthcare team can reduce moral distress amongst staff [66,67], prevent patients from feeling caught between opposing clinician views, and improve communication regarding treatment harm and misinterpretation of effects [43].

**CMO #8:** Prescribing inertia may be caused by a lack of communication or collaboration amongst HCPs (context). Specialized team-based collaborations may remove clinician concern for offending original prescribers or fear of making clinical errors (mechanism) leading to improved deprescribing efforts for patients in their last phase of life.

### Education, training and decision support for end-of-life shared decision-making

Interventions to improve clinical staff competencies in end-of-life SDM include didactic teaching, simulation communication training, algorithm support, reminder lists of potentially inappropriate medications (PIMs) and alterations to the electronic medical records to prompt clinician consideration for medication review and talking to patients about deprescribing. The Holmes model provides four considerations for deprescribing with patients with limited life expectancy: (1) treatment target; (2) time-until-benefit; (3) prognosis; and (4) goals of care [59]. Such decision supports has been combined with PIMS lists such as the Beers Criteria, the "STOPP" Screening Tool of Older People's Prescriptions, the "START" Screening Tool to Alert to Right Treatment, and the European-based Priscus list [68]. Integration of PIMS lists into electronic medical record (EMR) systems as a notation has also been recommended:

> "Beers Criteria are published as a list of medications by class alongside a recommendation; each recommendation has a rationale, a recommendation strength, and an evidence strength. Recommendations range from simply "Avoid" to "Use with caution" with caveats of specific instances that require avoidance or caution. The graphs are easy to use for the experienced or training clinician and could easily be implemented into an EMR as a notation in medication orders. [69] (p. 678).

Pop-up notifications and note templates in the electronic medical records may also be designed to remind or mandate clinicians to record the goals of care conversations they have had with patients. Evidence suggests that incorporating notifications in the electronic medical records ensures that SDM conversations are initiated by clinicians and the results of such SDM conversations are recorded for future review and use. This has shown to result in improved outcomes in line with accepted indicators of quality end-of-life care [56]. Additionally, electronic medical record notifications can include mandatory or voluntary form fields for recording SDM activity. Evidence suggests that optional form fields allow clinicians to respond to the pop-up notification and record notes when necessary and useful, and to ignore such notifications when not commensurate with real-time needs:

> "When signing a visit note, oncologists received a "soft alert" (that they can ignore) if the EHR lacked templated documentation of GOC discussion within the preceding 6 months. This alert aimed to enhance visibility and sustainability of the initiative without unduly burdening clinicians" [56] (p. 3401).

**CMO #9:** Prescribing inertia means clinicians avoid or do not remember to initiate medication review with patients on a regular basis (context). Scheduled pop-up notifications and note templates in the EMR can trigger reminders and logs for clinicians to engage in SDM with patients (mechanism) leading to improved communication with patients around goal concordance and deprescribing (outcome).

The evidence further suggests that stand-alone interventions such as providing clinicians with algorithm-based decision support using PIMs lists may not always be in line with clinical expertise and as such may create backfiring effects, such as junior clinicians avoiding medications on PIMs lists altogether [70]. Some have argued that results of algorithm support require advanced clinical appraisal skills:

*The authors mention a good palliative-geriatric guideline by Garfinkel and colleagues [71] that shows promise for cancer patients but the limitation of this guideline is that it requires a skilled physician to use the algorithm to guide the identification of PIMs. The algorithm is designed for palliative care patients and takes into account medications that provide limited benefit for this group such as statins and anti-hypertensives. However, some criteria are quite broad, and an experienced physician is necessary to use the guideline.* [30] *(p. 1117)*

On the other hand, it has been suggested that well designed real-time support tools reduce the need for combining multiple efforts, meaning labor-intensive efforts to educate healthcare providers on deprescribing can be more effectively accomplished by simply changing the electronic medical record (EMR) interface to include prompts. In a study to investigate if EMR modifications would be adequate without teaching sessions, the authors concluded that if the EMR modification is well designed, intuitive and comprehensive it is adequate as a stand-alone intervention to support end-of-life deprescribing [72]. Educational sessions which require that clinicians absorb information outside the clinical encounter, without real-time EMR modifications may be less effective due to 'alert fatigue' resulting in memory relapse [72] (p. 2).

Algorithm-based screening support tools may be contrasted with clinician expertise and intuition which may or may not align with the values presented in computational analysis of patient diagnosis. To bridge the potential gap between algorithm results and expertise-based clinical judgement, recent versions of the STOPPFrail criteria includes a 'surprise question':

*"The surprise question essentially functions as a method of separating those with an intermediate-to-high probability of dying (the clinician answers that he/she would not be surprised if the patient died within a year, i.e., surprize question positive). The authors note that recent systematic reviews have shown that the 'surprize question' does allow clinicians to exclude patients with longer survival times. This may be a safety net of patients not approaching the end of life, if there are consequences to removing medications too early* [42].

**CMO #10:** Clinicians are tasked with considering when would be the most appropriate time to deprescribe medications for patients approaching their last phase of life (context). The surprise question in screening tools asks clinicians to estimate the probability of the patient dying in the next 12 months. This embedded question reminds clinicians to balance the results of such applications with subjective, intuitive expertise (mechanism) thereby optimizing screening support for deprescribing (outcome)

Interventions have also been introduced for improving the psychological readiness and health literacy of patients and families. Patients with life-limiting conditions require adequate knowledge of medical treatment options and the ability to appraise different options in terms of relative benefit and harms. When clinicians do not provide adequate detail about benefits and harms of treatment options, patients may naturally defer decisions to clinicians:

*GPs should not assume that their patients have all the relevant information about their medications to enable them to participate in decision-making …GPs may interpret that older adults who are passive in consultations are lacking in health literacy skills. This…may deter them from initiating shared decision-making, instead assuming decision-making authority* [73] *(p. e22)*

On the patient side, interventions have been developed to improve the health literacy of patients and families including mobile applications, volitional help sheets, quizzes, and question prompt lists for preparing to engage in SDM with their healthcare team [74]. Question prompt lists have been given to patients before a clinical appointment to help them remember the questions they have; and realize questions they should be asking [75]. In another study, researchers used a simple questionnaire to determine the extent of goal concordance between patients and their healthcare team, suggesting that these data given to all parties can be used to simulate new conversations around goals of care. In that study, 798

**Table 4. Key principles for developing a shared decision support tool for ATT deprescribing.**

| |
|---|
| • A SDM tool should educate on and explain to clinicians, patients and carers about the problem with prescribing inertia and the importance of engaging in discussions about deprescribing with patients and families; |
| • The design of the tool should consider whether it will be used with clinicians, patients/families or all parties; |
| • The design should also consider whether the tool is meant to be used outside of the clinical encounter, during the doctor-patient visit, or both; |
| • The tool development should consider different options for decision support including algorithm-based, including PIMS list information, and activating patient self-interest and self efficacy. |
| • The tool design should also consider the opportunity to improve patient health literacy and information regarding their illness as well as tailor the information to the needs of patients given their illness progression. |
| • The tool should outline for patients, families and staff the reasons and value of initiating or engaging in discussions and sharing in decisions about prescribing or discontinuation of anti-thrombotic therapy, present options and information about harms/benefits associated with these options, and support patients to express their preferences |

patients were asked the following questions: "If you had to make a choice today, would you prefer (a) medical care that focuses on extending your life as much as possible, even if it means having more pain/discomfort; (b) medical care that focuses on relieving your pain and discomfort as much as possible, even if it means not living as long; or (c) I'm not sure. They were then asked, "which of the following best describes the type of medical care you are getting from your doctors right now?" with the same response options. Goal discordance was identified if they responded with different answers to the first versus second set of questions. The research demonstrated that when this patient's data were provided to staff, it improved staff motivation to increase advanced directive conversations and documentation by 98% [76].

**CMO #11:** Patients who have just received a terminal diagnosis may not be immediately ready to engage in SDM with clinicians responsible for their care (context). Support to prepare patients with common questions to ask HCPs and health literacy upskilling can motivate them to engage in conversations around deprescribing earlier and more frequently in their healthcare trajectory (mechanism), leading to improved deprescribing through and SDM approach in the last phase of life (outcome)

Given the complex interplay of factors involved in supporting efforts at SDM in this area, Tables 3 and 4 outline top-level findings and key principles for developing a shared decision support tool based on this review of the literature. Future decision support for ATT deprescribing for patients with cancer in the last phase of life should also refer to existing guidelines and standards for decision aid development given patient and clinical needs and perspectives [77,78].

## Discussion

This realist synthesis has explored literature on SDM and deprescribing in end-of-life care to build a conceptual platform for understanding how to optimize ATT management for persons living with cancer in the last year of life. The analysis covers three theory areas inductively derived from the evidence in the literature. These are: (1) the causes of prescribing inertia in the last year of life, (2) the mechanisms of SDM between clinician, patients and their families, and (3) educational and other supports to prepare for such SDM efforts. A main finding is that in circumstances of clinical uncertainty about risks and benefits, clinicians may avoid proactive decision-making on deprescribing and default to continuing existing therapies. In realist language, these circumstances leave mechanisms of deprescribing "inactivated." Profound clinical uncertainty in end-of-life care calls for better understanding of patient values and preferences for treatment should guide decision-making. However, while the efforts toward SDM may be supportive given the unknowns, evidence suggests the need for increased time in consultation, professional equipoise, fortitude to discuss end-of-life care, and advanced communication skills to explore what matters to people given the emotionally sensitive nature of end-of-life conversations. Thus, to support SDM in this context, specialized education, training, and decision support for all parties (clinicians, patients and families) is warranted.

In the specific case of ATT deprescribing, patients living with advanced cancer are often uninformed about, and surprised to experience cancer- and cancer therapy-induced venous thromboembolism [79,80]. They may be further shocked to experience severely distressing and disruptive bleeding events only to be informed after the fact that such events are attributable to their medications. Efforts to deprescribe ATT thus requires specialized considerations in relation to generic guidance on deprescribing. SDM with patients who have been on ATT for long-term conditions may require a different approach to communication as their understanding of, and attachment to, the medication will be different from those who were prescribed ATT at the time of cancer diagnosis. Furthermore, aligning ATT with existing, more established categorizations of potentially inappropriate medications for people living with cancer may help to trigger the right mechanisms for optional deprescribing. This categorization may include: (a) medications for long-term conditions which have been prescribed for many years (e.g., for blood pressure management, cholesterol, back injury; heart disease, depression etc.); (b) medications started at cancer diagnosis for curative intention including chemotherapy and radiation; (c) medication started at cancer diagnosis for prevention of secondary events, such as ATT. Doing so may better enable clinicians to explain the risks and benefits of deprescribing in a way that relates to patients' existing understanding. Furthermore, research has demonstrated that clinicians emphasizing the advantages of discontinuation over the risks of continuation can redirect patient and family attention to the importance of comfort and quality of life in the last phase of life [81].

Prior experience that raises the profile of ATT for individual patients (such as survivors of stroke or pulmonary embolism) may also play a role in their judgement on deprescribing. Similarly, deprescribing medications that patients associate with 'life saving or prolonging effects' such as chemotherapies, may require a specialized communication style that is also different from deprescribing medications that are perceived as less significant in terms of patient orientation to survivability and dying. For this reason, it is useful to consider how different medications and the reasons for their original prescribing impact how clinicians and patients feel in relation to deprescribing conversations, and to customize the approach to trigger anticipated deprescribing mechanisms optimally.

The findings of this review relate to other realist studies on deprescribing for older people, such as the TAILOR evidence synthesis [61], the deprescribing rainbow model [82] and MEMORABLE realist review [83], as well as Huisman and colleagues specific work on optimizing ATT prescribing at the end of life [9,10,81,84]. These studies have highlighted the importance of the provider-patient relationship in medication review, the need for flexibility and adaptation of deprescribing guidelines, the importance of organizational supports and understanding unique patient contexts. The SERENITY study adds to this growing body of research an understanding that deprescribing efforts can be thoughtfully framed in terms of SDM with patients and such clinician-patient conversations may optimize prescribing in the last phase of life. The findings of our review can also inform the development of de-prescribing practices from a focus on identification (of medicines and of people with limited life expectancy) to an approach that encompasses de-prescribing SDM in palliative care. Such approaches are not limited to people living with non-malignant conditions, as they are relevant to the palliative care of people living with respiratory, cardiac, and neurological conditions in community, hospice, and hospital settings [85–87].

ATT inhabits a grey area regarding patient understanding of the purpose of the medication and importance for deprescribing. Given that ATTs are not as high-profile as, for example, chemotherapies, clinicians soliciting patient input to achieve consensus on deprescribing may result in patient and family confusion about whey they are being asked, and clinicians may actively avoid initiating such discussions for their own emotional protection and self-image. A broader challenge in treating patients with cancer in the last year of life is the fact that clinicians may suppress their hunches about death prognosis to maintain a positive atmosphere for patients and staff [88]. This is one of the reasons why the START/STOPP decision tool asks clinicians a 'surprise question' (how surprised would you be if your patient died in the next 12 months?). Asking the question may indeed stimulate clinical thought patterns toward improving comfort care and away from proactive curative treatment, although in cancer care, 12 months from death may be too long to estimate death timing [89].

Organizational responses to prescribing and deprescribing should also be considered. This study corroborates other findings, in which various actors in a coordinated service may disagree on who should be prescribing and deprescribing

ATT [84,90,91]. For this reason, and supported by evidence, clinicians are advised to initiate SDM conversations early in the treatment journey, while the patient is still relatively healthy. Organizational culture and clinical leadership are essential elements to support this. Research has demonstrated that most patients are not upset by such conversations, and such efforts can help them orient toward full participation in decisions about their care as a terminal illness progresses. Clinicians need to feel comfortable providing their honest expert opinion to patients even when they are uncertain about the best course of action. Patients and families can benefit from being made aware that a clinician's expression of professional equipoise in the face of uncertainty is not a sign of resignation, but of holding a space of reflection to ensure optimal care and prevention of severely distressing and disruptive bleeding events in the short timeframe before death.

This realist synthesis has a number of important strengths. Given the dearth of studies specifically examining the role of SDM and deprescribing for ATT medications in the last year of life, and none (to our knowledge) exploring the contexts and mechanisms involved in implementation, the review sampled a broad range of studies conducted in high income countries, incorporated contributions from a wide range of stakeholders in Europe (including people with lived experience) and extrapolated lessons from the insight therein. However, although robust methods were used, underpinned by realist logic, important papers or other source material may have been omitted and we did not pursue additional routes of programme theory development to include knowledge from low and middle income countries. Although the exploratory connections were validated by patient, carer, and practitioner groups associated with the study, they remain tentative conclusions which will require further investigation and theory testing.

In conclusion, this realist synthesis has reviewed a mainly high-income country literature base on deprescribing and SDM in end-of-life care and has extrapolated mechanisms of action to better understand how to optimize ATT deprescribing for people with cancer in the last phase of life. Clinicians can benefit from understanding the context of prescribing decisions, how they are understood by the patient in the effort to advance discussions and improve SDM for this important consideration of ATT continuation or deprescribing. The benefit of the realist synthesis has been the opportunity to extrapolate from a diverse set of papers on deprescribing and SDM in end-of-life care, to the focussed question about the approach specifically addressing ATT deprescribing. Although there is no simple formula to be made for when and for whom ATT deprescribing is suitable, the findings presented here can inform the development of SDM tools to motivate all parties to engage in informed discussions about reviewing medications. Doing so can resolve the problem of prescribing inertia that causes devastating impacts on patients and their families as well as moral distress amongst healthcare staff.

## Supporting information

**S1 File. Initial Programme Theories.**
(DOCX)

**S2 File. Search narrative, search strategies and PRISMA-S checklist: Record and reasons for database searches and related decisions.**
(DOCX)

**S3 File. Details on stakeholder meetings with professionals and patients.**
(DOCX)

**S4 File. Included studies.**
(DOCX)

**S5 File. RAMESES Checklist.**
(DOCX)

## Acknowledgments

Disclaimer: Views and opinions expressed are however those of the authors only and do not necessarily reflect those of the European Union or The European Health and Digital Executive Agency.

## Author contributions

**Conceptualization:** Mark Pearson, Anthony Maraveyas, Fliss EM Murtagh, Simon Noble, Anette Arbjerg Højen, Frederikus A Klok, Miriam J Johnson.

**Formal analysis:** Justin Jagosh, Mark Pearson, Anthony Maraveyas, Miriam J Johnson.

**Funding acquisition:** Mark Pearson, Fliss EM Murtagh, Simon Noble, Frederikus A Klok, Miriam J Johnson.

**Investigation:** Justin Jagosh, Mark Pearson, Sarah Greenley, Anthony Maraveyas, Gamze Keser, Fliss EM Murtagh, Simon Noble, Michelle Edwards, Adrian Edwards, Anette Arbjerg Højen, Kathy Seddon, Frederikus A Klok, Simon P Mooijaart, Eric C.T. Geijteman, Miriam J Johnson.

**Methodology:** Justin Jagosh, Mark Pearson.

**Writing – original draft:** Justin Jagosh, Mark Pearson, Sarah Greenley, Miriam J Johnson.

**Writing – review & editing:** Justin Jagosh, Mark Pearson, Sarah Greenley, Anthony Maraveyas, Gamze Keser, Fliss EM Murtagh, Simon Noble, Michelle Edwards, Adrian Edwards, Anette Arbjerg Højen, Kathy Seddon, Frederikus A Klok, Simon P Mooijaart, Eric C.T. Geijteman, Miriam J Johnson.

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
