## [Editor Report · Decision Letter 0]

30 Sep 2024

Dear Dr Pearson, 

Thank you for submitting your manuscript entitled "How, when, and in which contexts should (dis)continuance of anti-thrombotic medicines be discussed with patients living with cancer in their last phase of life? A realist synthesis" for consideration by PLOS Medicine.

Your manuscript has now been evaluated by the PLOS Medicine editorial staff and I am writing to let you know that we would like to send your submission out for external peer review.

Please re-submit your manuscript within two working days, i.e. by Oct 02 2024 11:59PM. Please let us know if you need more time (ssunny@plos.org).

Kind regards,

Syba

Syba Sunny, MBBS, MRes, FRCPath

Associate Editor

PLOS Medicine

ssunny@plos.org

---

## [Decision Letter · Decision Letter 1]

20 Dec 2024

Dear Dr Pearson,

Many thanks for submitting your manuscript "How, when, and in which contexts should (dis)continuance of anti-thrombotic medicines be discussed with patients living with cancer in their last phase of life? A realist synthesis" (PMEDICINE-D-24-03264R1) to PLOS Medicine. The paper has been reviewed by four subject experts; their comments are included below and can also be accessed here: [LINK]

As you will see, the reviewers were positive about the study and the approach, but they raised several questions about the details of the methodology and the presentation. After discussing the paper with the editorial team and an academic editor with relevant expertise, I'm pleased to invite you to revise the paper in response to the reviewers' comments. We plan to send the revised paper to some or all of the original reviewers, and we cannot provide any guarantees at this stage regarding publication.

In view of the upcoming holidays, we ask that you submit your revised paper by Monday, January 6th. However, if this deadline is not feasible, please contact me by email (hvanepps@plos.org), and we can discuss a suitable alternative. Don't hesitate to contact me directly with any questions; otherwise, we look forward to receiving your revised manuscript in the new year. 

Kind regards,

Heather

Heather Van Epps, PhD

Executive Editor

[on behalf of]

Syba Sunny, MBBS, MRes, FRCPath 

Associate Editor

PLOS Medicine

ssunny@plos.org

Comments from the reviewers: 

Reviewer #1: 

Thank you for the opportunity to review a realist review titled 'How, when, and in which contexts should (dis)continuance of anti-thrombotic medicines be discussed with patients living with cancer in their last phase of life: A realist synthesis'. The review was conducted to support the development of SDM support tools for those involved in ATT at the end of life related to cancer, as part of a HORIZON research project.

After reading the manuscript I feel its title needs adjustment, given the research questions and the discussed literature mainly cover deprescribing and SDM at the end of life in general. The method section states, 'the analysis focussed on locating causal insights in the published literature regarding deprescribing for patients in the last phase of life and shared decision-making in that context' (without specifying what that context may be). There is some focus on ATT but the program theory closely resembles those developed by other realist syntheses around deprescribing and SDM without explaining ATT specific contexts and mechanisms (some nuances are mentioned in the discussion). While it will be applicable to and support the development of SDM support tools for discontinuation of ATT it seems not reflective of the chosen title.

Methodology:

1. L 116 - impede not impeded?

2. L 116 - refining theories - as theory development has not yet been mentioned, 'refining' seems out of context here.

3. L 123 - 'illuminating (that word always sounds a bit like AI) the importance of contextual factors' doesn't quite align with the end of the sentence 'and reasons why it can fail' which in a realist sense would usually describe mechanisms.

4. L 145 - 'The main search for literature was developed and conducted' - this sound as if the information specialist started afresh, did they use/ build on the initial scoping searches, adapting them to each database? I am curious why so many of them were searched, wouldn't it be highly likely that the approach to selection, RCQ and expert solicitation, would have yielded the most relevant papers by searching the databases most relevant to the field of investigation?

5. L 177 - throughout it is not clear whether evidence for shared decision making AND deprescribing was evaluated, OR either or. The search strategy was formulated with OR, but shared decision making to achieve deprescribing is often a particularly difficult process to establish. At this point in the manuscript, I was under the assumption the synthesis tries to establish Cs and Ms to achieve deprescribing of ATT as a shared decision.

6. Given the complexity of the topic and overall search it may be difficult to outlay it with exacting clarity (without asking readers to go to appendix 3) but it wasn't clear why some terms were included without reference to deprescribing, e.g. moral distress - although the chosen papers relate to end-of-life care, in the manuscript it isn't clear that this was the case. It may be worthwhile providing a clearer picture and rationale of how the searches (and later choices of included papers) relate to each other, in addition to the listing of the stepwise approach taken.

7. It may also be beneficial at this point and in the findings to state clearly that due to the sparsity of literature dealing specifically with deprescribing of ATT in the patient group of interest focus had to be adjusted and extrapolation took place. That would probably avoid confusion among some readers (mentioned below).

8. Overall, it may be easier for readers to follow the program theory statements if they were placed after the narrative summary of the literature. This would answer a number of questions I had when reading through them following the current order.

9. L 227 - it would be good to outline how the areas of prescribing inertia and training for decision support were established. For example, the first is a well-established concept in the general deprescribing literature, was it only drawn from the papers included in the synthesis?

10. L 237 - It would be helpful for the if - then statements to be clearly signposted as they are not configured as CMOCs (as described in the methods), the 'then' at times stands in for the 'because' of a mechanism. Which ones relate to contexts, which ones to mechanisms? CMOCs may not be necessary as the outcome for all would be deprescribing of ATT? 

11. Some statements seem to describe either a context or a mechanism and then an outcome, which again was confusing in terms of realist logic. For example, 1c seems to describe a problem of mechanisms, clinicians feeling uncertain etc. leading to an outcome of persistent inertia. As a context it probably would be better formulated as the fact that there is uncertainty about risk-benefit. In the absence of evidence-based guidance clinicians negotiate unmeasurable harms and benefits (rather than 'feeling' uncertain there is real uncertainty) which leads to the maintenance of the status quo, or inertia.

12. For the domain of shared decision making, it would also be good to discern that 2a and b relate to mechanisms whereas statements 3 a and b describe training / education and the access to decision support as enabling contexts for those Ms.

Narrative summary:

13. L 391 & 405 I am not sure that the mention of one paper supporting a concept warrants the use of the term 'literature'.

14. L 417 The statement of 'Shared decision-making across interprofessional staff teams has been shown to counter prescribing inertia' would need a reference as the literature (the way it is described) within the following paragraph does not support or evidence it.

15. L 454 While the Beers or STOPP/START criteria may assist in making decisions on deprescribing for older people in general, they are not tailored to end-of-life decisions in cancer patients (many of whom are not old). It is not clear whether they are discussed as general examples of the presence of decision support tools achieving deprescribing at a higher rate than their absence or as useful in making decisions to deprescribe ATT in cancer patients.

16. L 544 This flows on into table 3. I thought this realist review was supposed to underpin SDM support tools for the deprescribing of ATT in cancer patients, whereas the recommendations in the table seem to refer to something more generic. While many of them will be exactly the same for something more specific, what would adding a PIMS list contribute when targeting a very specific group of medicines?

Discussion: 

17. I found the discussion difficult to follow, partly because some sentences are very long, e.g. L 557-561, 558-592 (breaking them up would assist readers), and because it seems to repeat some of the findings but also discuss new ones. It reads a bit like everyone may have been adding points they thought needed mentioning, which interrupts cohesion and makes it quite long for a realist review, in which the details of the program theory would be the main discussion points. 

18. It also reflects what I perceive as the shifting between general deprescribing at the end of life and that of ATT throughout the manuscript. It raises issues and arguments which I would have expected in the findings. For example, I wonder whether the discussion points at L 578 (if supported by literature) shouldn't be included in the PT, as ATT as primary or secondary prophylaxis or treatment of thromboembolic events and whether anticoagulants or antiplatelet agents are in use would greatly influence clinicians' and people's reasoning around its (de)prescribing. As such the statements starting in L 623, that the review addresses the complexity of the architecture of deprescribing of a specific class of medications, seems a stretch too far, particularly as conclusions have been reached mainly by extrapolation rather than evidence.

19. For transparency it would be good to add the reference numbers to the papers listed in appendix three. It is frustrating trying to look up what has been numbered within the manuscript but not the table when wanting to find out which paper is discussed.

Reviewer #2: 

Thank you for the opportunity to review this manuscript, which represents a mixed methods study, with realist synthesis methodology, including international literature searches (database, expert recommendation), and international stakeholder engagement (clinicians from different professions and specialties, and people with lived experience) to explore desprescribing practices in relation to ATTs in the context of malignancy.

I agree that this is an important topic, which spans multiple specialities, and thus likely to be of interest to the broad readership of PLOS Medicine. I suggest some minor revisions which might strengthen the paper. I am not an expert in realist synthesis, but if the Editor or other reviewer are, then it would be helpful to ensure that methodology is appropriately described and adhered to.

1. Abstract: I thought the potential mechanisms outlined underpinning SDM were helpful and could be considered/trialled by clinicians and managers. In fact, I think these should also be explicitly included in the main text- otherwise, if I search e.g. 'empowerment' or 'investment' in the main text, these don't show up at all, and I have to wade through the Discussion to see where the authors mentioned these issues.

Intro: 

2. Good overview of issue. 

3. Paragraph 2 first few points would benefit from some citations to back up statements (even though intuitively I agree). There is some literature around deprescribing towards end of life and in other terminal conditions (e.g. dementia) which may be of relevance here.

4. The boxes were helpful for clarity.

Methods: 

5. "RCQ is a screening strategy in which papers are screened starting from the most recent date and working in reverse chronology until a quota is filled." Please include planned a priori quota.

6. Please include rough numbers for consumer groups included in "The other two meetings were held through a patient and public involvement group, comprising people living with, and supporting people with cancer, as well as members of the public at large."

Results:

7. "Four key content areas organised the dataset." Can the authors mention how these were determined- from which part of the described methodological process?

8. Figures are helpful for clarity. Small pint: please give legend for abbreviations with the figure (e.g. PIMS, eMR for Figure 2- although I know what these are, some readers may not and they are defined later in the text from current siting of Figure 2.

9. "The findings * are organised across three areas (prescribing inertia, shared decision-making, 228 education, training and decision support for shared decision-making) and seven sub-areas 229 (Figure 2)." *Pleas add 'of the literature review' if I have interpreted correct.

10. For each of the thematic areas 1-3 (pages 11-19 in Reviewer copy), I think it would be useful to divide the test into 'problem's and 'potential solutions' as these are 2 aspects of the same theme.

11. Re SDM tool: "A SDM tool should educate on and explain to clinicians the problem with prescribing inertia and the importance of engaging in discussions about deprescribing with patients and families" Shouldn't it also educate patients/carers? It should be used by the clinician and patient/carer together, so both will need education on its use/what it can achieve.

Discussion:

12. "The analysis covers three theory areas": if these were developed from the analysis, and not pre-determined, this should be highlighted.

13. "The findings of this review relate to other realist studies on deprescribing such as the TAILOR 606 evidence synthesis [57] the deprescribing rainbow model [77] and MEMORABLE realist review 607 [78],"- It would be good if the patient cohorts included in each of these could be bracketed, for reader's easy reference.

14. Please see above re summary findings in Abstract, not mirrored explicitly in main text.

Reviewer #3: 

Thanks so much for this excellent review of deprescribing. Though it is targeted about antithrombotic drugs it delves deeply into related contextual issues of deprescribing in general, shared decision-making and EMR-generated triggers to encourage advance care planning and goals of care discussions ("Serious Illness Conversations"). My only minor criticism is that the title focusing on ATT does not do it full justice. With good key words I hope however that it will be easily findable when searching general terms.

Pippa Hawley

Reviewer #4:

Thank you for the opportunity to review this manuscript. This is a realist synthesis on how when and what kind of settings should clinicians discuss about continuing or discontinuing anti thrombotic medication. This is a relatively unique setting for the application of the realist synthesis because it has not been well applied yet in clinical decision-making settings such as this. However, the methodology lends itself well to the setting being studied. Shared decision-making between clinicians and caregivers or patients is a very important area for application of realist methodology because of various considerations at the individual, patient and caregiver characteristics level, at the history and nature of relationship between the caregiver and provider, as well as various workplace and other environmental considerations within the clinical settings where such conversations occur. Given the multilevel nature of these interactions shared decision making cannot be studied merely from the angle of effectiveness of the educational content provided by the clinician, but rather needs to take into consideration multilevel interactions across various domains which indeed span the discipline of psychology, communication, and organisational sciences. Such interdisciplinary approaches could help clinicians better understand what kind of shared decision making approaches work for which patients and in what settings and for the latter, the realist approach. 

My comments here in should be interpreted from a place of expertise in the methodology. I am not an expert in the clinical area being studied here, although I am a trained clinician myself and continue a small portion of my time on clinical care.

I compliment the author team for carefully bringing the strengths of the realist approach to this area of clinical enquiry in shared decision-making. Indeed, I believe the overall approach they have taken for studying shared decision making in the context of discontinuation of ATT in end of life Cancer care can be expanded to be applied for shared decision-making conversations in various other clinical settings as well. I find the application of the realist approach to have been done using state of the Art guidance relevant for this. The development of if - then propositions which explain the phenomenon being studied and form a hypothetical basis for refining using observation and quality of data is well presented.

The authors do not substantively discuss the geographical and social settings that are represented in the 35 papers that they have retained. This has two important implications. One is related to relative lack of literature from Limited resource across the World, predominantly in LMIC settings, but also in Limited resource settings in high income countries as well. The second concern is the fact that a lot of social science, theorising is done in high income and well-resourced settings which includes a probable bias in the program theories and mechanisms that arise out of such work. Indeed, this bias has been well pointed out in literature as well. Discussing the kind of evidence that they have identified and which inform the review (and what kind of evidence that they may have expected that was NOT found in the evidence-base), and critically sharing in the discussion section/limitations section what implication this has on their findings is something useful to consider and is offered here as an optional suggestion to the authors. If this is not feasible, it may be important to qualify their assertion in the conclusion that their synthesis is based on an "…international literature base" (the question being how international is the evidence-base?).

---

* Please upload any figures associated with your paper as individual TIF or EPS files with 300dpi resolution at resubmission; please read our figure guidelines for more information on our requirements: http://journals.plos.org/plosmedicine/s/figures. While revising your submission, please upload your figure files to the PACE digital diagnostic tool, https://pacev2.apexcovantage.com/. PACE helps ensure that figures meet PLOS requirements. To use PACE, you must first register as a user. Then, login and navigate to the UPLOAD tab, where you will find detailed instructions on how to use the tool. If you encounter any issues or have any questions when using PACE, please email us at PLOSMedicine@plos.org.

* Please ensure that the study is reported according to the RAMESES guideline (https://www.equator-network.org/reporting-guidelines/rameses-publication-standards-realist-syntheses/) and include the completed RAMESES checklist as Supporting Information. When completing the checklist, please use section and paragraph numbers, rather than page numbers. Please add the following statement, or similar, to the Methods: "This study is reported as per RAMESES guideline (S1 Checklist)."

FIGURES AND TABLES

SUPPLEMENTARY MATERIAL

REFERENCES

---

## [Decision Letter · Decision Letter 2]

29 Apr 2025

Dear Dr. Pearson,

Thank you very much for re-submitting your manuscript "How can current evidence on shared decision-making and deprescribing in palliative care support anti-thrombotic therapy (dis)continuance for persons living with cancer in their last phase of life? A realist synthesis" (PMEDICINE-D-24-03264R2) for review by PLOS Medicine.

Thank you for your detailed response to the reviewers' and editors’ comments. I have discussed the paper with my colleagues, and it has also been seen again by three of the original reviewers. The changes made to the paper were mostly satisfactory to the reviewers. As such, we intend to accept the paper for publication, pending your attention to the reviewers' and editors' comments below in a further revision. When submitting your revised paper, please once again include a detailed point-by-point response to the reviewers' and editorial comments.

[LINK]

In revising the manuscript for further consideration here, please ensure you address the specific points made by each reviewer and the editors. In your rebuttal letter you should indicate your response to the reviewers' and editors' comments and the changes you have made in the manuscript. Please submit a clean version of the paper as the main article file. A version with changes marked must also be uploaded as a marked up manuscript file. Please also check the guidelines for revised papers at http://journals.plos.org/plosmedicine/s/revising-your-manuscript for any that apply to your paper. 

We ask that you submit your revision within 1 week (May 06 2025). However, if this deadline is not feasible, please contact me by email, and we can discuss a suitable alternative.

Please do not hesitate to contact me directly with any questions (atosun@plos.org). If you reply directly to this message, please be sure to 'Reply All' so your message comes directly to my inbox.

We look forward to receiving the revised manuscript.

Sincerely,

Alexandra Tosun, PhD

Associate Editor

PLOS Medicine

plosmedicine.org

Comments from Reviewers:

Reviewer #1: Thanks to the authors for addressing the reviewers' comments and clarifications/changes within the manuscript. The literature search/review section now is easier to follow and the discussion more concise.

Reviewer #2: Thank you to the authors for addressing the Reviewer comments from multiple reviewers, including my own.

I feel that these have been satisfactorily answered, and where not, rationale has been given.

I did think that the bracketed point by Reviewer 4 "Discussing the kind of evidence that they have identified and which inform 

the review (and what kind of evidence that they may have expected that was NOT found in the evidence-base)" would have enriched the Discussion (without adding overly to Word Count), but this is a small point.

I see the authors have deferred to the Editorial team regarding the need to address two points specifically raised by Reviewers, and I likewise defer to Editorial team for same:

-Reviewer 1, point 19 on pg 4 of Response, regarding Table 4 references

-Reviewer 2, point 10 on Response pg 6, regarding problems/potential solutions 

Reviewer #4: The revised version has adequately responded to my review and concerns raised by other reviewers.

[LINK]

Requests from Editors:

GENERAL

* Please confirm that your title complies with to PLOS Medicine's style. Your title must be nondeclarative and not a question. It should begin with main concept if possible. "Effect of" should be used only if causality can be inferred, i.e., for an RCT. Please place the study design ("A randomized controlled trial," "A retrospective study," "A modelling study," etc.) in the subtitle (i.e., after a colon).

Suggestion: Shared decision-making and deprescribing in palliative care as support for anti-thrombotic therapy (dis)continuance for persons living with cancer in their last phase of life: A realist synthesis

* Please revise for use of patient-centered language. Please note that patient-centered language is constructed with the use of post-modified nouns (e.g. 'patients with cancer’ (or similar) instead of ‘cancer patients’) putting the person first in the sentence structure.

* Please ensure that all abbreviations are defined at first use throughout the text (including statistical abbreviations). Please also check figures and tables.

* Please review your text for claims of novelty or primacy (e.g. 'for the first time', ‘novel’) and remove this language.

* Please ensure that tables and figures, including those in supplementary files, are appropriately referenced in the main text.

* We appreciate that "family" is considered an important factor in the synthesis. Have you given any thought or consideration to individuals who don't have family support, may have a legal guardian, etc.?

* ll.748-751: We believe these are competing interest. Please make sure to include these in the relevant section in the online submission form.

* If possible, we suggest presenting how many of the studies included details on either of one of the three main topics identified. 

* Thank you for providing the RAMSES checklist. Please replace the page numbers with paragraph numbers per section (e.g. "Methods, paragraph 1"), since the page numbers of the final published paper may be different from the page numbers in the current manuscript.

* Regarding the points raised by Reviewer #2, we strongly encourage you to reconsider both of these points. We think that it should be clear which statements describe either a context or a mechanism and then an outcome, and that dividing the text into "problems" and "potential solutions" might be a useful structure for the text. However, we will not make these two points editorial requirements.

ABSTRACT

* Please confirm that your abstract complies with our requirements, including providing all the information relevant to this study type https://journals.plos.org/plosmedicine/s/submission-guidelines#loc-abstract

* l.39, “Realist synthesis examining context-mechanism interactions.” – please change to a full sentence.

* l.43: Please list the databases.

* We think it would be useful if you could outline the types of articles covered by the 56 papers and 35 papers identified (i.e. how many were qualitative, how many were mixed methods, etc.).

* l.47: What does " genuine " mean in this context? Is this a scientific description? Please avoid emotive language.

* l.61: “actual, rather than feared” – Is this from a quote or is this your personal description?

* How many citations were eventually included in the synthesis?

* Please outline the inclusion/exclusion criteria.

* Please ensure that all numbers presented in the abstract are present and identical to numbers presented in the main manuscript text.

* l.68, we suggest changing to “that may causes devastating”.

AUTHOR SUMMARY

* Please refer to low or middle income countries rather than "developing countries" or "the Global South". Please refer to high income countries rather than "developed" or "Global North" countries. Please revise throughout the main text.

* If you agree, under "What did the researchers do and find?" we think you could add one or two bullet points explaining the theories in more detail.

INTRODUCTION

* Please ensure that the Introduction ends with a clear description of the study question or hypothesis.

* ll.132-134: We think that the review questions (Table 1) together with the definitions of study terms (Table 2) should be moved to the Methods section.

METHODS AND RESULTS 

* When revising the manuscript, please keep in mind that PLOS Medicine has a broad audience that may not be familiar with the methodology of a realist synthesis. Please ensure that you provide a clear, structured, and easy-to-follow methodology that outlines the key steps in performing a realist synthesis.

Examples from PLOS One and PLOS Global Public Health: 

https://doi.org/10.1371/journal.pgph.0003330

https://doi.org/10.1371/journal.pone.0285222

* In line with the comment above, we think that the structure of the Methods section could be improved, e.g. by subheadings (e.g. according to the sections outlined in the RAMESES checklist).

* l.152: Please remove claims of novelty or primacy.

* l.170, “widen the main search” – please explain in more detail and ensure to add a relevant reference to the full search terms. 

* l.175: Was there a set of certain exclusion criteria? Based on which criteria were the 180 citations excluded? Please clarify.

* Figure 1: We suggest modifying Figure 1 to reflect (parts of) the PRIMSA flow diagram which also includes details on the databases searched, exclusion criteria etc. 

* ll.195-204: These two sections seem rather disconnected from the main text. As you can see from a previous comment, we feel that the mention of inclusion and exclusion criteria should be part of the main text. Please revise.

* l.210: “Members of the SERENITY research team and consortium recommended papers which were also included. – this statement is repetitive, please remove.

* l.210, “quota is filled” – what was the quota?

* l.250: Please note that on line 250, there’s a ‘RESULTS’ heading while on 279 you added a ‘Findings’ heading. Please remove the latter.

* l.290, “catastrophic” – please consider whether the use of strong emotive language is necessary here. We suggest removing it.

* ll.292-303: We suggest transforming the text into a table.

* Figure 1: Please ensure to define all abbreviations used and to explain numbers 1-3.

* l.310: Please remove the numbering. You may use “The “Prescribing Inertia” Problem in End-of-Life Care” as a subheading. This also applies to the subsequent subheadings.

* l.389: Please note that you should consistently use the abbreviation "SDM" for shared decision making once you have introduced it. Please revise throughout.

* Table 4: Have you referenced Table 4 in the text?

DISCUSSION

* Please remove all subheadings.

* We think the issues/topics you are discussing are important beyond anti-thrombotic Rx in terms of deprescribing at the end of life. We were wondering if you could be a little more explicit about the general interest of the question beyond anti-thrombotics for patients with cancer?

General Editorial Requests

---

## [Editor Report · Decision Letter 3]

20 Jun 2025

Dear Dr Pearson, 

On behalf of my colleagues and the Guest Academic Editor, Karen Luetsch, I am pleased to inform you that we have agreed to publish your manuscript "Shared decision-making and deprescribing to support anti-thrombotic therapy (dis)continuance for persons living with cancer in their last phase of life: A realist synthesis" (PMEDICINE-D-24-03264R3) in PLOS Medicine.

I appreciate your thorough responses to the reviewers' and editors' comments throughout the editorial process. We look forward to publishing your manuscript, and editorially there are only a few remaining points that should be addressed prior to publication. We will carefully check whether the changes have been made. If you have any questions or concerns regarding these final requests, please feel free to contact me at atosun@plos.org.

Please see below the minor points that we request you respond to:

1) Checklist: Please provide sections and paragraph, such as "Methods, paragraph 1".

2) Author Summary: Please ensure that the formatting of the last bullet point is correct. In the track changes version, the last bullet point is missing. In the PDF, the last bullet point ends with "Introduction," which we assume is meant to be the first heading for the main text.

3) Please ensure to upload Figure 1 as a separate file.

4) Please check to see if you introduced the abbreviation "CMOc" (we believe you only introduced "CMO").

5) Discussion: Please remove "Strengths and Limitations" and "Conclusion" from the last two paragraphs, as they serve as a type of subheading. Please turn them into full sentences.

Before your manuscript can be formally accepted you will need to complete some formatting changes, which you will receive in a follow up email (including the editorial points above). Please be aware that it may take several days for you to receive this email; during this time no action is required by you. Once you have received these formatting requests, please note that your manuscript will not be scheduled for publication until you have made the required changes.

PRESS

Sincerely, 

Alexandra Tosun, PhD 

Senior Editor 

PLOS Medicine